# Learning on Arbitrary Graph Topologies via Predictive Coding

**Tommaso Salvatori**[1,*]  **Luca Pinchetti**[1,*]  **Beren Millidge**[2]  **Yuhang Song**[1,2,†]
**Tianyi Bao**[1]  **Rafal Bogacz**[2]  **Thomas Lukasiewicz**[3,1]

[1] Department of Computer Science, University of Oxford, UK
[2] MRC Brain Network Dynamics Unit, University of Oxford, UK
[3] Institute of Logic and Computation, TU Wien, Austria

`tommaso.salvatori@cs.ox.ac.uk, luca.pinchetti@cs.ox.ac.uk`
`beren.millidge@ndcn.ox.ac.uk, yuhang.song@some.ox.ac.uk, tianyi.bao@cs.ox.ac.uk`
`rafal.bogacz@ndcn.ox.ac.uk, thomas.lukasiewicz@cs.ox.ac.uk`

## Abstract

Training with backpropagation (BP) in standard deep learning consists of two main steps: a forward pass that maps a data point to its prediction, and a backward pass that propagates the error of this prediction back through the network. This process is highly effective when the goal is to minimize a specific objective function. However, it does not allow training on networks with cyclic or backward connections. This is an obstacle to reaching brain-like capabilities, as the highly complex heterarchical structure of the neural connections in the neocortex are potentially fundamental for its effectiveness. In this paper, we show how predictive coding (PC), a theory of information processing in the cortex, can be used to perform inference and learning on arbitrary graph topologies. We experimentally show how this formulation, called *PC graphs*, can be used to flexibly perform different tasks with the same network by simply stimulating specific neurons. This enables the model to be queried on stimuli with different structures, such as partial images, images with labels, or images without labels. We conclude by investigating how the topology of the graph influences the final performance, and comparing against simple baselines trained with BP.

## 1  Introduction

Classical deep learning has achieved remarkable results by training deep neural networks to minimize an objective function. Here, every weight parameter gets updated to minimize this function using reverse differentiation [1, 2]. However, in the brain, every synaptic connection is independently updated to correct the behaviour of its post-synaptic neuron [3] using local information, and it is unknown whether this process minimizes a global objective function. The brain maintains an internal model of the world, which constantly generates predictions of external stimuli. When the predictions differ from reality, the brain immediately corrects this error (difference between reality and prediction) by updating the strengths of the synaptic connections [4–7]. This theory of information processing, called *predictive coding (PC)*, is highly influential, despite experimental evidence in the cortex being mixed [8–11], and it is at the centre of a large amount of research in computational neuroscience [12–16]. From the machine learning perspective, PC has promising properties: it is able to achieve excellent results in classification [17–19] and memorization [20, 21], and is able to process information in both a bottom up and a top down direction. This last property is fundamental

---

† Corresponding author.
* Equal contribution.

36th Conference on Neural Information Processing Systems (NeurIPS 2022).

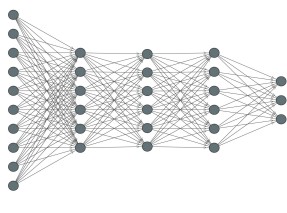 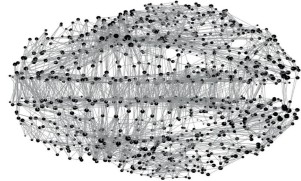

Artificial Neural Network                    Biological Neural Network

Figure 1: Difference in topology between an artificial neural network (left), and a sketch of a network of structural connections that link distinct neural elements in a brain (right) [26].

for the functioning of different brain areas, such as the hippocampus [22, 20]. PC also shares the generalization capabilities of standard deep learning, as it is able to approximate backpropagation (BP) on any neural structure [23], and a variation of PC is able to exactly replicate the weight update of BP on any computational graph [24, 25]. Moreover, PC only uses local information to update synapses, allowing the network to be fully parallelized, and to train on networks with any topology.

Training on networks of any structure is not possible in standard deep learning, where information only flows in one direction via the feedforward pass, and then BP is performed in sequential steps backwards. If a cycle is present inside the computational graph of an artificial neural network (ANN), BP becomes stuck in an infinite loop. More generally, the computational graph of any function $F: \mathbb{R}^d \to \mathbb{R}^k$ is a poset, and hence acyclic. While the problem of training on some specific cyclic structures has been partially addressed using BP through time [27] on sequential data, the restriction to hierarchical architectures may present a limitation to reaching brain-like intelligence, since the human brain has an extremely complex and entangled neural structure that is heterarchically organized with small-world connections [26]—a topology that is likely highly optimized by evolution. This shape of structural brain networks, shown in Fig. 1, generates a unique communication dynamics that is fundamental for information processing in the brain, as different aspects of network topology imply different communication mechanisms, and hence perform different tasks [26]. The heterarchical topology of brain networks has motivated research that aims to develop learning methods on graphs of any topology. A popular example is the *assembly calculus* [28, 29], a Hebbian learning method that can perform different operations implicated in cognitive phenomena.

In this work, we address this problem by proposing *PC graphs*, a structure that allows to train on any directed graph using the original (error-driven) framework by Rao and Ballard [7]. We then demonstrate the flexibility of such networks by testing the same network on different tasks, which can be interpreted as conditional expectations on different neurons of the network. Our PC graphs framework enables the model to be queried on stimuli with different structures, such as partial images, images with labels, or images without labels. This is significantly more flexible than the strict input-output structure of standard ANNs, which are limited to scenarios when they are always presented with data and labels in the same format.

Note that the main goal of this work is not to propose a specific architecture that achieves state-of-the-art (SOTA) results on a particular task, but to present PC graphs as a new flexible and biologically plausible model, which can achieve good results on many tasks simultaneously. In this work, we study the simultaneous generation, classification, and associative memory capabilities of PC graphs, highlighting their flexibility and theoretical advantages over standard baselines. Our contributions are briefly summarized as follows:

- We present PC graphs, which generalize PC to arbitrary graph topologies, and show how a single model can be queried in multiple ways to solve different tasks by simply altering the values of specific nodes, without the need for retraining when switching between tasks. Particularly, we define two different techniques, which we call *query by conditioning* and *query by initialization*.

- We then experimentally show this in the most general case, i.e., for fully connected PC graphs. Here, we train different models on MNIST and FashionMNIST, and show how the two queries can be used to perform different generation tasks. Then, we test the model on classification tasks, and explore its capabilities as an associative memory model.

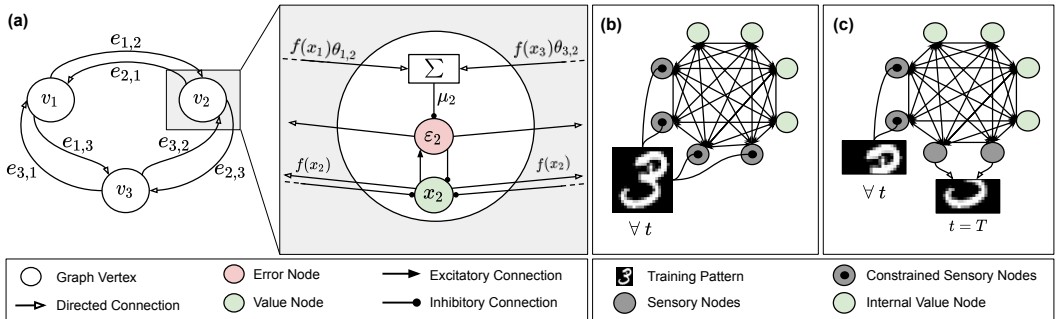

Figure 2: (a) An example of a fully connected PC graph with three vertices. Zoomed is the neural implementation of PC, where learning is made local via the demonstrated inhibitory and excitatory connections. (b) A sketch of the training process, where the value nodes of the sensory vertices are fixed to the pixels of the image. (c) A sketch of query by conditioning, where a fraction of the value nodes is fixed to the top half of an image, and the bottom half is recovered via inference.

- We next investigate how different graph topologies influence the performance of PC graphs on generation tasks, reproducing common network architectures such as feedforward, recurrent, and residual networks as special cases of PC graphs, and investigate how the chosen structure influences the performance on generative tasks. Finally, we also show how PC graphs can be used to derive the popular *assembly calculus* [28].

## 2 PC Graphs

Let $G = (V, E)$ be a directed graph, where $V$ is a set of $n$ vertices $\{1, 2, \ldots, n\}$, and $E \subseteq V \times V$ is a set of directed edges between them, where every edge $(i, j) \in E$ has a weight parameter $\theta_{i,j}$. The set of vertices $V$ is partitioned into two subsets, the *sensory* and *internal vertices*. External stimuli are always presented to the network via sensory vertices, which we consider to be the first $d$ vertices of the graph, with $d < n$. The internal vertices, on the other hand, are used to represent the internal structure of the dataset. Each vertex $i$ encodes several quantities. The main quantity is given by the values of its activity, which change over time, and we refer to it as a *value node* $x_{i,t}$. We call the value nodes of the sensory vertices *sensory nodes*. Additionally, each vertex computes the *prediction* $\mu_{i,t}$ of its activity based on its input from value nodes of other vertices:

$$\mu_{i,t} = \sum_j \theta_{j,i} f(x_{j,t}), \tag{1}$$

where the summation is over all the vertices $j$ connected to $i$ via outgoing edges, and $f$ is a non-linearity. Equivalently, it is possible to consider the summation on every $j$, and have $\theta_{i,j} = 0$ if $(i, j) \notin E$. The error of every vertex at every time step $t$ is then given by the difference between its value node and its prediction, i.e., $\varepsilon_{i,t} = x_{i,t} - \mu_{i,t}$. This local definition of error, which lies not only in the output, but in every vertex of the network, is what allows PC graphs to learn using only local information. The value nodes $x_{i,t}$ and the weight parameters $\theta_{i,j}$ are updated to minimize the following energy function defined locally on every vertex:

$$\mathcal{E}_t = \frac{1}{2} \sum_i (\varepsilon_{i,t})^2. \tag{2}$$

A fully connected PC graph with 3 vertices is sketched in Fig. 2a, along with the operations that describe the dynamics of the information flow, showing also how every operation can be represented via inhibitory and excitatory connections.

**Learning:** When presented with a training point $\bar{s}$ taken from a training set, the value nodes of the sensory vertices are fixed to be equal to the entries of $\bar{s}$ for the whole duration of the training process, i.e., for every $t$. A sketch of this is shown in Fig. 2b. Then, the total energy of Eq. (2) is minimized in two phases: *inference* and *weight update*. During the inference phase, the weights are fixed, and the value nodes are continuously updated via gradient descent for $T$ iterations, where $T$ is a hyperparameter of the model. The update rule is the following (*inference*):

$$\Delta x_{i,t} = -\gamma \cdot \partial \mathcal{E}_t / \partial x_{i,t} = \gamma \cdot (-\varepsilon_{i,t} + f'(x_{i,t}) \sum_{k=1}^n \varepsilon_{k,t} \theta_{i,k}), \tag{3}$$

where $\gamma$ is the learning rate of the value nodes. This process of iteratively updating the value nodes distributes the output error throughout the PC graph. When the inference phase is completed, the value nodes get fixed, and a single weight update is performed as follows (*weight update*):

$$\Delta\theta_{i,j} = -\alpha \cdot \partial\mathcal{E}_T/\partial\theta_{i,j} = \alpha \cdot \varepsilon^l_{i,T} f(x_{j,T}), \quad (4)$$

where $\alpha$ is the learning rate of the weight update. We now describe two different ways to query the internal representation of a trained model, where the values of some sensory vertices are undefined, and have to be predicted. In both cases, the weight parameters $\theta_{i,j}$ are now fixed, and the total energy $E$ is continuously minimized using gradient descent on the re-initialized value nodes via Eq. (3).

**Query by conditioning:** While each value node is randomly re-initialized, the value nodes of specific vertices are *fixed* to some desired value, and hence not allowed to change during the energy minimization process. The unconstrained sensory vertices will then converge to the minimum of the energy given the fixed vertices, thus computing the conditional expectation of the latent vertices given the observed stimulus. Formally, let $I = \{i_1, \ldots, i_q\} \subset \{1, 2, \ldots, n\}$ be a strict subset of vertices. Assume now that we know that a subset of the value nodes corresponding to the vertices $I$ is equal to a stimulus $\bar{q} \in \mathbb{R}^q$. Then, running inference until convergence allows to estimate the conditional expectation

$$E(\bar{x}_T \mid \forall t\colon (x_{i_1,t}, \ldots, x_{i_q,t}) = \bar{q}), \quad (5)$$

where $\bar{x}_T$ is the vector of value nodes at convergence. Examples of tasks performed this way are (i) classification, where internal nodes are fixed to the pixels of an image, and the sensory nodes are fixed to a 1-hot vector with the labels, (ii) generation, where the single value node encoding the class information is fixed, and the value nodes of the sensory nodes converge to an image of that class, and (iii) reconstruction, such as image completion, where a fraction of the sensory nodes are fixed to the available pixels of an image, and the remaining ones converge to a reasonable completion of it. A sketch of this process is shown in Fig. 2c.

**Query by initialization:** Again, every value node is randomly initialized, but the value nodes of specific nodes are *initialized* (for $t = 0$ only), but not fixed (for all $t$), to some desired value. This differs from the previous query, as here every value node is unconstrained, and hence free to change during inference. The sensory vertices will then converge to the minimum found by gradient descent, when provided with that specific initialization. Again, let $I = \{i_1, \ldots, i_q\} \subset \{1, 2, \ldots, n\}$ be a strict subset of vertices, and assume that we have an initial stimulus $\bar{q} \in \mathbb{R}^q$. Then, we can estimate the conditional expectation

$$E(\bar{x}_T \mid (x_{i_1,0}, \ldots, x_{i_q,0}) = \bar{q}). \quad (6)$$

Examples of tasks performed this way are (i) denoising, such as image denoising, where the sensory neurons are initialized with a noisy version of an image, which is cleared during the energy minimization process, and (ii) reconstruction, such as image completion, where the fraction of missing pixels is now not known a priori.

## 3 Proof-of-concept: Experiments on Fully Connected PC Graphs

In this section, we perform experiments on a fully connected PC graph $G = (V, E)$, i.e., where $E = V \times V$. Such PC graphs are fully general and encode no implicit priors on the structure of the dataset. It is possible to obtain any possible graph topology by simply pruning specific weights of $G$.

Given a dataset of $m$ datapoints $\mathcal{D} = \{\bar{s}_i\}_{i<m}$, with $\bar{s}_i \in \mathbb{R}^d$, we train the PC graph as described in Section 2: The first $d$ neurons are fixed to the entries of a training point, and the energy function $\mathcal{E}_t$ is minimized via inference and weight updates, via Eqs. (3) and (4). When the training is complete, we show the different tasks that can be performed, without the need of retraining the model. We use MNIST and FashionMNIST [30], fixing the first $d$ nodes to the data point, and show how to perform the tasks of generation, denoising, reconstruction (without and with labels), and classification by querying the PC graph as described in Section 2.

**Setup:** For every dataset, we have trained 3 models: one for generation and classification tasks, one for denoising and reconstructions, and one for associative memories. The first two models consist of a fully connected graph with 2000 vertices, trained with 794 sensory vertices for classification and generation tasks (784 pixels plus a 1-hot vector for the 10 labels), and 784 sensory vertices

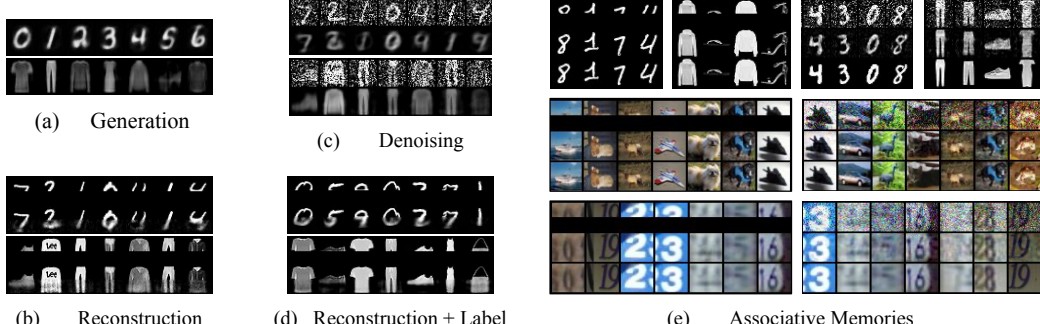

|            |                    |            |                      |                      |
|------------|--------------------|------------|----------------------|----------------------|
| (a)        | Generation         | (c)        | Denoising            |                      |
| (b)        | Reconstruction     | (d)        | Reconstruction + Label | (e)   Associative Memories |

Figure 3: Generation experiments using the first 6 classes of the MNIST and FashionMNIST datasets from the labels {0, 1, 2, 3, 4, 5, 6} and {t-shirt, trouser, pullover, dress, coat, sandal, shirt}, respectively; (b) reconstruction of incomplete images using *query by conditioning*, when only the top half is available; (c) reconstruction of corrupted images using *query by initialization*; (d) reconstruction of incomplete images using *query by conditioning* when also providing the correct label of the test image; and (e) associative memory experiments when presented with half of a training image (left) or a corrupted version (right) that it has already seen and memorized; from top to bottom row: image provided to the network, retrieved image, and original image.

for reconstruction and denoising. Further details about other hyperparameters are given in the supplementary material.

**Generation:** To check the generation capabilities of a trained PC graph, we queried the model by conditioning on the labels: Here, the value nodes dedicated to the 10 labels were fixed to each 1-hot value, and the energy of the model (Eq. (2)) was minimized using Eq. (3) until convergence. The generated images are then taken to be the value nodes of the unconstrained sensory nodes, which were originally fixed to the pixels of the images during training. An example of the images generated for each label is given in Fig. 3a.

**Reconstruction:** We provide the PC graph with half of a test image, and ask it to reconstruct the second half. This can be done using both queries: when querying by conditioning, half of the pixels of a test image are fixed to the corresponding sensory nodes; when querying by initialization, the value nodes are simply initialized to the same values. At convergence, we consider the value nodes of the unconstrained nodes, which should reconstruct the missing part of the image based on the information learned during training. The results are given in Fig. 3b. We have also replicated the same experiment using a network trained with the labels, and provided the label during the reconstruction. This computes the distribution of the missing pixels knowing the available ones *and* the label. The results in this case are visibly better and are given in Fig. 3d.

**Denoising:** We provide the PC graph with a corrupted image, obtained by adding zero-mean Gaussian noise with variance 0.5. This is done by querying by initialization: before running inference, the value nodes of the sensory nodes are initialized to be equal to the pixels of the corrupted image. At convergence, we consider the value nodes of the unconstrained nodes, which should reconstruct the original image. The results are given in Fig. 3c.

**Results:** As stated above, we picked a fully connected PC graph due to its generality, and not to obtain the best performance. However, the results show that this framework is able to learn an internal representation of a dataset, and that it can be queried to solve multiple tasks with a reasonable accuracy. The PC graph was in fact able to always generate the correct digit, and almost always able to generate the correct clothing item in generation tasks, and always able to provide a noisy but reasonable reconstruction of incomplete test points. The same happened with denoising experiments, as a cleaner (plausible) image was always produced. In Section 4, we show how to improve all these performances by using different PC graph topologies.

**Classification:** We consider the same PC graph trained for the generation experiments. To check its generalization capabilities, we query by conditioning the pixels of every test image to the first 784 sensory nodes, and run inference to reconstruct the 1-hot label vector. We do not expect to obtain results directly comparable with standard multilayer perceptrons for two reasons: firstly, the

Table 1: Test accuracy of different models on MNIST, FashionMNIST, and SVHN.

| Model | Ours | Hopfield Nets | Boltzmann Machine | Almeida Pineda |
|---|---|---|---|---|
| MNIST | $91.76 \pm 0.02$ % | $65.23 \pm 2.21$ % | $79.23 \pm 0.15$ % | $76.36 \pm 0.14$ % |
| FashionMNIST | $83.72 \pm 0.33$ % | $51.74 \pm 3.94$ % | $61.31 \pm 0.17$ % | $69.63 \pm 1.64$ % |
| SVHN | $84.51 \pm 0.11$ % | $48.92 \pm 3.11$ % | $55.74 \pm 1.23$ % | $59.14 \pm 2.64$ % |

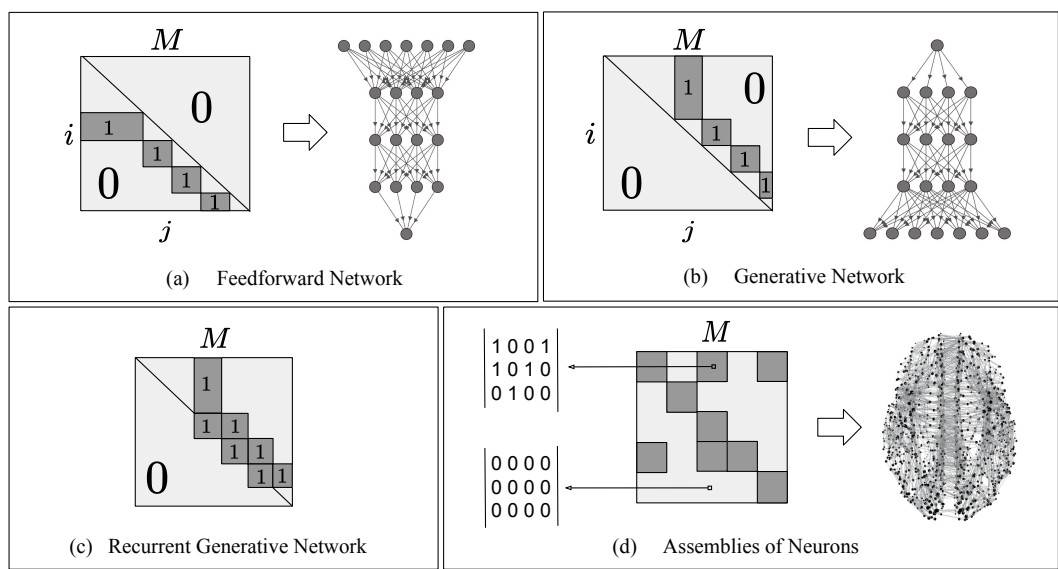

Figure 4: Examples of PC graphs that can be built by masking a part of the weights of a fully connected PC graph. (a) Masking required to build a standard multilayer architecture, such as the one in [17]. (b) Masking required to build a multilayer architecture, where the weights go in the opposite direction. Here, the sensory nodes are at the end of the hierarchical structure. This model is equivalent to the generative networks in [20]. (c) Examples of masking needed to implement popular architectures with lateral connections, similar to the model in [35]. (d) This is the model in [28], which consists of a set of Erdõs–Renyi graphs that simulate brain regions (dark squares on the diagonal) and connections between them (dark squares off the diagonal).

model does not contain any implicit hierarchy, which empirically appears crucial to obtaining good classification results. Secondly, the PC graph is also simultaneously learning to generate the pixels, which are much more numerous than labels. However, to check whether the obtained results were acceptable, we tested against different learning algorithms that train on similar or equivalent fully connected architectures, such as Hopfield networks, unconstrained Boltzmann machines, and a local variation of BP introduced in the late '80, called Almeida-Pineda, named after the two scientists who independently invented it [31, 32]. As for Hopfield networks, we used the implementation provided in [33]. The results, given in Table 1, show that our model outperforms every other learning algorithm that can be trained on fully connected architectures. Despite this, the results also show that the obtained test accuracy is not nearly comparable to the results obtained by multilayer perceptrons, as they are only slightly better than a linear classifier (obtaining $88\%$ accuracy on MNIST). However, this is not due to the learning rule of PC, which is well-known to be able to reach a competitive performance when provided with a hierarchical multilayer structure [17]. For the SVHN [34] experiment, we used models with 5000 vertices.

**Associative memory:** We now test whether PC graphs are able to memorize training images and retrieve them given a corrupted or incomplete version of it. Particularly, we show that a fully connected PC graph is able to store complex data points, such as colored images, and retrieve them via running inference. To do that, we trained a novel fully connected PC graph on 100 data points of the MNIST, FashionMNIST, CIFAR10, and SVHN datasets. We have used a model with 1000 vertices for MNIST and FashionMNIST, and 3500 for SVHN and CIFAR10, and asked it to retrieve

the original memories by presenting it either only half of the original pixels, or a corrupted version with Gaussian noise variance 0.2. This task is similar to image reconstruction and denoising, with the non-trivial difference that here we only use already seen data points, and hence no generalization is involved. The results of these experiments are given in Fig. 3e, and show that our method is able to successfully store and retrieve data points via energy minimization. More details about the capacity of fully connected PC graphs are given in the supplementary material.

## 4   Extension to Different PC Graph Topologies

As well-known in deep learning, the performance of the trained model strongly depends on its architecture: the number of vertices, layers, and their intrinsic structure. In Section 3, we studied the general architecture of fully connected PC graphs. Here, we show how to reduce a fully connected PC graph to lighter and even more powerful PC graphs. Particularly, we show how to generate different neural architectures by simply pruning specific edges of a fully connected PC graph $G = (V, E)$. In this case, the pruning is performed by applying a sparse mask $M$. However, there are multiple equivalent ways of implementing it. Consider now the weight matrix $\bar{\theta} \in \mathbb{R}^{n \times n}$, where every entry $\theta_{i,j}$ represents the weight parameter connecting vertex $i$ to vertex $j$. To generate a neural architecture that consists of a subset of the original connections, it suffices to *mask* the matrix $\bar{\theta}$ via entry-wise multiplication with a binary matrix $M$, where $M_{i,j} = 1$ if the edge $(i, j)$ exists in $E$, and $M_{i,j} = 0$ otherwise. This allows the creation of hierarchical discriminative architectures such as a PC equivalent of the multilayer perceptron (MLP) in Fig. 4a, or hierarchical generative networks in Fig. 4b, c. More generally, it creates a framework to generate and study architectures with any topology, such as small-world networks inspired by brain regions [36], as shown in Fig. 4d. Guidance on which topology should be used depends on the tasks and dataset, and it is hence hard to propose a general theory (as it is with BP). In what follows, however, we provide multiple examples.

**Experiments:** Here, we study how the network topology influences the final performance, performing the same experiments shown on the fully connected PC graph. We expect the generated images to be visibly better due to the enforced hierarchical structure of the PC graph.

**Setup:** We trained generative PC graphs, recurrent generative PC graphs, assemblies of neurons PC graphs, and standard BP autoencoders with different numbers of hidden layers and hidden dimension, and report the best results. For the generation results, we used the same setup, but added an input layer with 10 vertices, whose value nodes during training were initialized with the 1-hot label vector. We performed a search across learning rates $\gamma$ and $\alpha$, and on the number of iterations per batch $T$. More details are given in the supplementary material, as well as a long discussion on how different parameters influence the final performance of the architecture.

**Results:** The results are given in Fig. 5a and b. As expected, the hierarchical structure of the considered PC graphs improves over the fully connected PC graph, despite being comparable in the number of parameters. Compared against autoencoders (Fig. 5c), the standard ANN baseline trained with BP, the PC graph results are similar in image denoising, and better in image reconstruction. FID scores on denoising tasks for different levels of noise are given in Table 7.

## 5   Conditioning on Labels

Assume that we need to reconstruct a test image from an incomplete version of it, with the further assumption that that this time we are also provided with the label of the corrupted image. It would be useful to be able use this extra information to obtain a better reconstruction. In PC graphs, this is straightforward: it suffices to simultaneously fix the value nodes representing the labels to the 1-hot vector of the provided label, and the sensory nodes to the pixels of the corrupted image. This method can be applied when it is difficult to infer to which class an incomplete image belongs, and providing the label during the reconstruction allows the preferred label to influence the reconstruction. Hence, we perform the following task: we provide images of digits that look similar when incomplete, and ask the model to reconstruct the missing half when giving the label information, i.e., use the additional label information to correctly resolve the inherent ambiguity in the reconstruction task.

**Experiments:** We used the same PC graphs from above for generation tasks. We provided the PC graph the bottom $33\%$ of random images representing 7s or 9s. Note that it is hard to distinguish between these two numbers when only this small portion of the image is available. Then, we generated

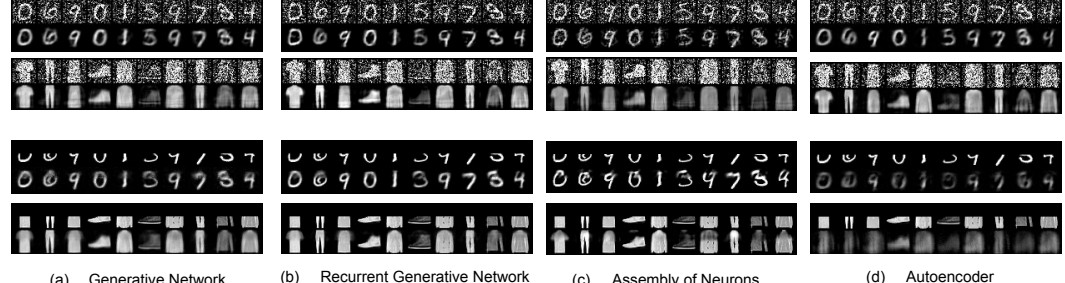

|     (a)   Generative Network | (b)   Recurrent Generative Network | (c)   Assembly of Neurons | (d)   Autoencoder |

Figure 5: *Query by initialization* (top) and *query by conditioning* (bottom) on three different PC graph architectures and different datasets. Particularly, we tested these PC graphs against ANN autoencoders trained with BP (d), which perform comparably to the PC graphs on denoising tasks, but less well on image reconstruction.

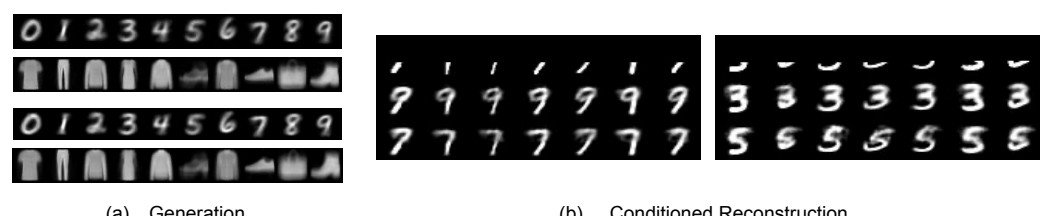

(a)   Generation                    (b)   Conditioned Reconstruction

Figure 6: Left: Generated images given the labels using feedforward (top) and recurrent (bottom) PC graphs. Right: conditional inference on the labels.

the missing $67\%$ of the pixels by first giving 7 as a label, and then giving 9. We have repeated the same task using 3s and 5s. The results, available in Fig 6b, show that our model is able to perform conditional inference, as the reconstructed digits always agree with the provided labels.

## 6   Assembly of Neurons

Recently, a model made by assemblies of neurons that are sparsely connected with each other has been proposed to emulate brain regions [28]. This model consists of $m$ ordered clusters of neurons $(C_1, \ldots, C_m)$, and any two ordered neurons of the same cluster are connected by a synapse with probability $p$, creating an Erdõs–Renyi graph $G_{m,p}$. Depending on the desired task, two clusters can be connected via sparse connections following the same rule: if cluster $C_a$ is connected to cluster $C_b$, then, given a vertex $v_i \in C_a$ and a vertex $v_j \in C_b$, there exists a synaptic connection connecting $v_i$ to $v_j$ with probability $p$. Note that this structure is highly general, and allows to build networks such as the one represented in Fig. 1b. To conclude, at each time step, only the $k$ neurons of every cluster with the highest neural activity fire. In the original work, the authors propose a Hebbian-like learning algorithm, however, we show that it can also be trained using PC graphs. A graphical representation on how to encode as a PC graph a network made by assemblies of neurons is given in Fig. 4d. In this case, each dark block on the diagonal represents connections between neurons of the same region. Unlike the other networks in the same figure, these are sparse matrices where every entry is either zero, or one with probability $p$. As in the brain, not every region is connected with the other, and whether two regions are directly connected has to be decided a priori when designing the architecture. Again, two neurons between connected regions are directly connected with probability $p$. In Fig. 4d, dark blocks off the diagonal represent the presence of directed connections between two regions $C_a$ and $C_b$. If situated below the diagonal, the connections go from $C_a$ to $C_b$, with $a < b$; if situated above the diagonal, they go from $C_b$ to $C_a$.

Figure 7: FID Score on MNIST on images corrupted with Gaussian noise of different variance.

| Method | PC | Autoencoder |
|--------|-------|-------------|
| 0.2 | 25.61 | 43.93 |
| 0.5 | 44.53 | 53.79 |
| 0.7 | 51.38 | 57.56 |

**Experiments:** We replicated this structure, using $4$ clusters with $3000$ vertices each, connected in a feedforward way: the first cluster is connected with the second, which is connected with the third, and so on. As sparsity and top-k constants, we used $p = 0.1$ and $k = 0.2$, and performed the same generative experiments. The results are given in Fig. 5c. While the results look cleaner than the other methods, note that they are specific to MNIST and FashionMNIST, as the top-k activation on the last cluster well cleans the noise surrounding the reconstructions.

# 7 Related Work

Our work shares similarities and the final goal with a whole field of research that aims to improve current neural networks by using techniques from computational neuroscience. In fact, the biological implausibility and limitations of BP highlighted in [37, 38] have fueled research in finding a new learning algorithm to train ANNs, with the most promising candidates being energy-based models such as *equilibrium propagation* [39, 40]. Other interesting energy-based methods are Boltzmann machines [41–43], and Hopfield networks [44, 45]. These differ from PC, as they do not encode the concept of *error*, but learn in a pure Hebbian fashion. Furthermore, they have undirected synaptic connections, and make predictions by minimizing a physical system initialized with a specific input. This is different from PC, that has directed synaptic connections and is tested by fixing specific nodes to an input, while letting the latent ones converge. The PC literature ranges from psychology to neuroscience and machine learning. Particularly, it offers a single mechanism that accounts for diverse perceptual phenomena observed in the brain, examples of which are end-stopping [7], repetition-suppression [46], illusory motions [47, 48], bistable perception [49, 50], and even attentional modulation of neural activity [51, 52], and it has even been used to describe the retrieval and storage of memories in the human memory system [22].

Although inspired by neuroscience models of the cortex, the computational model introduced by Rao and Ballard [7] still presents some implausibilities, with the main one being the presence of symmetric connections. An implementation of PC with no symmetric connections that is able to successfully learn image classification tasks has been presented in [53], and in the *neural generative coding* models, used for continual learning, generative models, and reinforcement learning [54, 55].

# 8 Discussion

In this work, we have shown that PC is able to perform machine learning tasks on graphs of any topology, called PC graphs. Particularly, we have highlighted two main differences between our framework and standard deep learning: flexibility in structure and query. On the one hand, a flexible structure allows for learning on any graph topology, hence including both classical deep learning models, and small-world networks that resemble sparse brain regions. On the other hand, flexible querying allows the model to be trained and tested on data points that carry different kinds of information: supervised signals, unsupervised, and incomplete. On a much broader level, this work strengthens the connection between the machine learning and the neuroscience communities, as it underlines the importance of PC in both areas, both as a highly plausible algorithm to train brain-inspired architectures, and as an approach to solve corresponding problems in machine intelligence.

The research of this paper (and current PC literature in general) is also of great importance from another perspective: training modern neural networks with BP has become computationally extremely expensive, making modern technologies inaccessible. Biological neural networks, on the other hand, do not have these drawbacks thanks to their biological hardware. Recent breakthroughs in the development of neuromorphic and analog computing, such as the finding of the *missing memristor* [56], could allow the training of deep neural models using only a tiny fraction of energy and time that modern GPUs need. To do this, however, we need to train neural networks end-to-end on the same chip, something that is not possible using BP (or BP through time), due to the need of a control signal that passes information between different layers. The energy formulation of neuroscience-inspired models allows to overcome this limitation, making them perfect candidates to train deep neural models end-to-end on the same chip [57]. This strongly motivates research in PC and other neuroscience-inspired algorithm, with a potentially huge long-term impact.

## Acknowledgments

This work was supported by the Alan Turing Institute under the EPSRC grant EP/N510129/1, by the AXA Research Fund, the EPSRC grant EP/R013667/1, the MRC grant MC_UU_00003/1, the BBSRC grant BB/S006338/1, and by the EU TAILOR grant. We also acknowledge the use of the EPSRC-funded Tier 2 facility JADE (EP/P020275/1) and GPU computing support by Scan Computers International Ltd. Yuhang Song was supported by the China Scholarship Council under the State Scholarship Fund and by a J.P. Morgan AI Research Fellowship.

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
