**Algorithm 1** Learning the external stimulus $\bar{s}$

---

**Require:** $(x_{1,t}, \ldots, x_{d,t})$ is fixed to $(s_1, \ldots, s_d)$.
 1: **for** $t = 0$ to $T$ **do**
 2:     **for** each vertex $i$ **do**
 3:         update $x_{i,t}$ to minimize $\mathcal{E}_t$ via Eq. (3)
 4:     **end for**
 5:     **if** $t = T$ **then**
 6:         update every $\theta_{i,j}$ to minimize $\mathcal{E}_t$ via Eq. (4).
 7:     **end if**
 8: **end for**;

---

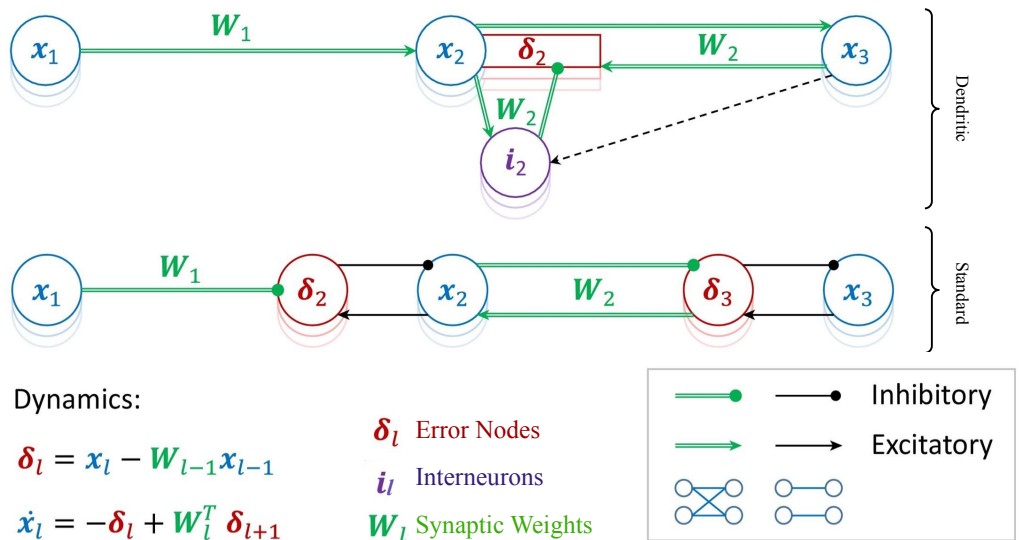

Figure 8: Standard and dendritic neural implementation of predictive coding. The dendritic implementation makes use of interneurons $i_l = W_l x_l$ (according to the notation used in the figure). Both implementations have the same equations for all the updates, and are hence equivalent; however, dendrites allow a neural implementation that does not take error nodes into account, improving the biological plausibility of the model. Figure taken and adapted from [38].

## A  A Discussion on Biological Plausibility

In the literature, there is often a disagreement on when a specific algorithm can be considered biologically plausible. This follows, as every computer simulation fails to be completely equivalent to every aspect on how the brain works, as there will always be some details that make the simulation implausible. Hence, it is normally assumed that an algorithm is biologically plausible when it satisfies a list of properties that are also satisfied in the brain. Different works consider different properties. In our case, we consider as list of minimal properties that a learning rule should satisfy, the ones that allow to have a possible neural implementation, such as local computations and lack of a global control signal to trigger the operations. However, the neural implementation proposed in Fig. 2 takes error nodes into account, often considered implausible from the biological perspective [58]. Even so, the biological plausibility of our model is not affected by this: it is in fact possible to map PCNs on a different neural architecture, in which errors are encoded in apical dendrites rather than separate neurons [58, 38]. Graphical representations of the differences between the two implementations is given in Fig. 8, taken (and adapted) from [38].

# B  Methodology and Further Experiments

Compared to backpropagation (BP), predictive coding (PC) allows for more flexibility in the definition, training, and evaluation of the model. The experiments reported in this paper show the best results achieved on each specific task and, as a consequence, only the effects of a specific set of hyperparameters. Therefore, the complete range of possibilities that exist in PC has not been displayed, however, those alternative configurations may be helpful in other scenarios. A pseudocode that describes the learning process of PC graphs is given in Algorithm 1.

## B.1  Architectures and Hyperparameters

In this section, we provide a detailed description of the models and parameters used to obtain the results in the various generation tasks presented in this work, to guarantee their reproducibility. Note that our goal was to compare the performance of different models, hence we compare networks that have a similar number of parameters. We now briefly summarize the PC graphs used in this work:

- **Fully connected networks:** The experiments in the paper body are obtained by using a fully connected graph with 2000 vertices, trained with 794 sensory vertices for classification and generation tasks (784 pixels plus a 1-hot vector for the 10 labels), and 784 sensory vertices for reconstruction and denoising. For colored images, we used a network with 5000 vertices. We trained every model for 20 epochs, and reported the best results using early stopping. As learning rates, we used $\alpha \in \{1, 0.5\}$ for the value nodes, and $\eta \in \{0.0001, 0.00005\}$ for the weights, and a weight decay $\lambda = \{0.01, 0.001, 0.0001, 0\}$. To conclude, we computed each query using $T = 2000$, making sure that the energy had converged before reaching that value.

- **Feedforward network:** A network composed by a sequence of $L$ fully connected layers of dimension $H$. The best results were achieved with $L \in \{3, 4\}$ and $H = 512$ for MNIST and $H = 1024$ on FashionMNIST. We did not experience any benefits in adding extra layers, as it only resulted in higher convergence times. The width, instead, directly determines the quality of the images produced: as expected, very narrow networks fail to store enough information to accurately reconstruct (or denoise) the input images. However, wide networks manifest sub-optimal performance as well. This follows, as having more parameters allows the network to easily overfit. As a consequence, the generation process is less stable, and the images can appear noisier and composed by strokes belonging to different classes. Using a strong weight decay alleviates these problems, as we will later discuss.

- **Recurrent network:** A recurrent layer consists of a layer whose output is transformed by a non-linear transformation and fed in input to the layer. The recurrent networks used in this paper consist of two recurrent layers (for a total of four non-linear transformations) with hidden dimension $H = 512$ when trained on MNIST, and $H = 1024$ when trained on FashionMNIST. The behaviour, given the choice of width and depth, seems similar to feedforward networks. The performance, however, seems to be less impacted by the usage of wide layers. This is due to the recurrent connections that establish more constraints, and thus stability.

- **Assembly of neurons:** As stated in the paper body, we used models with 4 clusters with 3000 vertices each, connected in a feedforward way. As sparsity and top-k constants, we used $p = 0.1$ and $k = 0.2$, and performed the same generative experiments. Again, we trained each model for 20 epochs, and reported the best results using early stopping. As learning rates, we used $\alpha \in \{1, 0.5\}$ for the value nodes, and $\eta \in \{0.0001, 0.00005\}$ for the weights. To conclude, we computed each query using $T = 2000$, making sure that the energy had converged before reaching that value.

- **Autoencoders:** The autoencoder was defined using the same shape as the feedforward networks: it is as a fully connected network with $L \in \{3, 4\}$ hidden layers of width $H \in \{256, 512, 1024\}$. In this way, the structure and the number of parameters directly correspond to the feedforward network trained using predictive coding. It was trained through BP using the *Adam* optimizer, with learning rate $\alpha = 1e^{-4}$ and weight decay of parameter $\lambda \in \{1e^{-2}, 1e^{-4}, 1e^{-6}, 0\}$ (the best results were achieved with the lowest value).

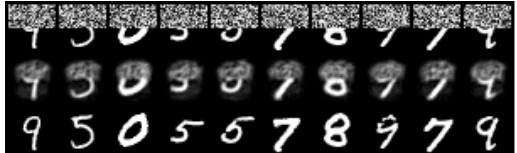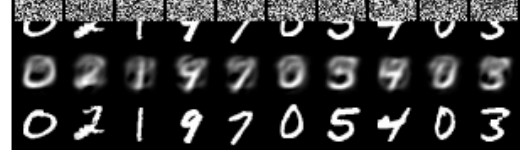

Figure 9: Reconstruction using query by conditioning on the whole output layer. The performance of feedforward networks (left) is noticeably improved by using recurrent connections (right), as the reconstructed images do not overfit the noise, but resemble plausible, albeit noisy, digits.

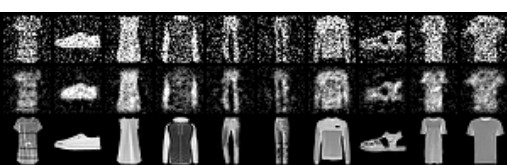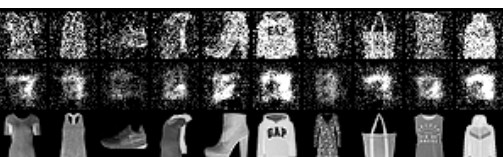

Figure 10: Reconstruction using query by conditioning using FashionMNIST samples after training on MNIST. Feedforward networks (left) simply overfit (i.e., reproduce without performing any modification) the input samples, despite being unrelated to the training data. Recurrent networks, instead, reproduce an unrecognisable and shady image, showing that they do not recognize the input samples, as they are not stable data points.

As predictive coding requires two sets of updatable parameters, the value nodes $x_{i,t}$ and the weights $\theta_{i,j}$, we defined two separate optimizers. The learning rate for the weights was set to $\alpha = 1e^{-4}$, and the optimizer algorithm chosen was *Adam* (as for the autoencoder). We experimented with different values of weight decays, noticing how the final performance is highly affected by this value. For the given tasks, the best results were achieved with *weight decay* $= 1e^{-2}$. Instead, the learning rate for the value nodes was set to $\gamma = 1.0$, and optimized using *SGD*. To conclude, we have tested different activation functions; the most promising seems to be *HardTanh*.

## B.2 Feedforward vs. Recursive Networks

In this work, we highlighted how in different situations, one may prefer to query by conditioning or by initialization. As a rule of thumb, conditioning means that we expect the partial data given to the network to be correct and be recognized as a *memory*, by being reconstructed by the network without modifications. Therefore, it makes sense to use it in the reconstruction generative task. Instead, when performing image denoising, we do not want the network to recall the noisy image from its memory, instead, we are asking it to retrieve the memory (or to generate a realistic sample), representing a plausible image, that is the closest to the noisy input. It makes therefore sense to only initialize the output layer, giving the network a direction to follow and let it evolve unconstrained. However, it may not always be clear which querying technique is most preferable. A desirable behavior may be using the network to identify which querying data are realistic (i.e., similar to the training samples) and which not. Ideally, we would like the network to perfectly fit previously seen data points, while struggling to reconstruct unfamiliar shapes. We tested both the feedforward and recursive networks by training them on the MNIST dataset and querying them by conditioning the output layer with a full-size image composed by half uniform noise and half digit. The results are reported in Fig. 9. We can see how feedforward networks easily fit the noise, reconstructing the two halves independently. On the other hand, employing recurrent connections (and thus imposing stricter constraints) forces the network to reconstruct the image as a whole. We can see a similar behavior in Fig. 10, where networks trained on MNIST are use to denoise FashionMNIST images. Feedforward networks easily overfit the input samples. Recurrent networks, instead, correctly do not recognize the given images and reconstruct an unrelated and confused blob. In this last case, it would therefore be possible to distinguish between familiar and unfamiliar images by computing the distance between the input and output images.

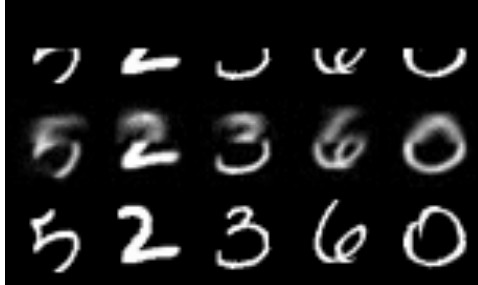 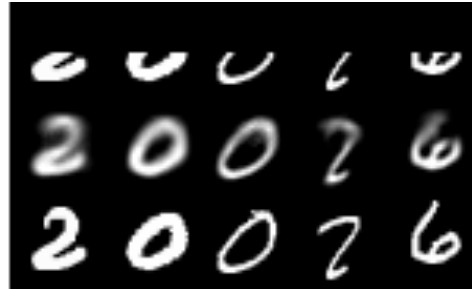

Figure 11: Reconstructed images given the label and by conditioning the bottom half. Using low weight decay values (left) causes the two halves of the images to be uncorrelated. As a result, each digit is composed by almost unrelated lines. Contrarily, with higher values (right), each image is correctly generated.

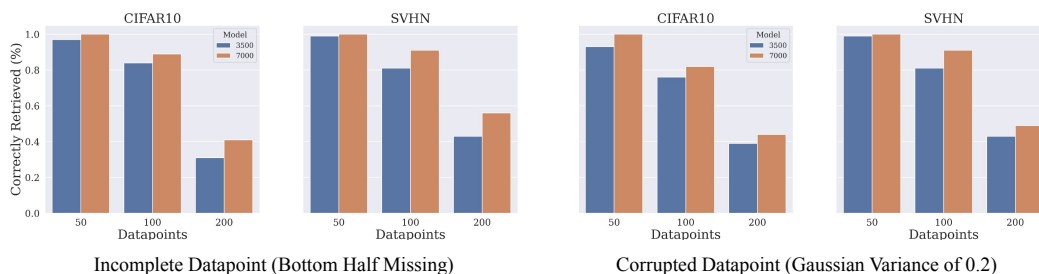

Figure 12: Number of correctly retrieved data points when presented with incomplete or corrupted variations. We used datasets of $\{50, 100, 200\}$ images of the CIFAR10 and SVHN datasets, and trained on fully connected PC graphs of size $\{3500, 7000\}$ vertices.

### B.3  Importance of Weight Decay

As previously mentioned, weight decay plays a fundamental role in determining the properties of the reconstructed images. Compared to other tasks (e.g., classification) or models (e.g., autoencoder trained by BP), a higher value of weight decay seems to be necessary when training with PC. From our experiments, weight decay prevents the networks from overlearning the task that they are trained on (i.e., reproduce any image that they are given in input), and instead allows them to "understand" the several concept classes of each dataset. This behaviour makes it possible to generalize their knowledge to new and unseen tasks, such as the denoising and reconstructing tasks seen in this work. It is worth noticing how, when optimizing for a single specific problem (e.g., image recognition), lower values of weight decay seem to be more effective.

To show this, we trained a recurrent network to reconstruct images by conditioning the bottom half of the output layer and giving the target class label in input. The result is that, with low weight decay, the network treats each half of the image independently, reconstructing the bottom part by fitting the conditioning data and the top half using the given label. It can be observed that there is no relation between the two halves. With higher weight decay, instead, we can see that the image is reconstructed as a whole, incorporating both the information provided via the label and the conditioning data (Fig. 11).

## C  Associative Memory Experiments

In the paper body, we claimed that a fully connected PC graph is able to perform associative memory (AM) experiments. To show this, we trained fully connected PC graphs with $\{3500, 7000\}$ vertices on different subsets of cardinality $\{50, 100, 200\}$ of CIFAR10 and FashionMNIST. Then, we used query by initialization and conditioning to retrieve the original memories. In this setting, we considered a memory to be retrieved if the mean squared error between the original training point and its

| Model | PC | RBM | DAM | BP |
|---|---|---|---|---|
| MNIST | $98.47 \pm 0.12$ | $94.12 \pm 0.59$ | $98.58$ | $98.41 \pm 0.18$ |
| FashionMNIST | $89.92 \pm 0.23$ | $86.98 \pm 0.49$ | $90.22 \pm 0.27$ | $90.29 \pm 0.33$ |
| SVHN | $88.99 \pm 0.26$ | $85.09 \pm 0.87$ | $86.77 \pm 0.22$ | $89.31 \pm 0.09$ |
| CIFAR10 | $56.23 \pm 3.36$ | $41.12 \pm 3.88$ | $46.06 \pm 2.77$ | $59.11 \pm 2.47$ |

Table 2: Test accuracy of multilayer PCNs (i.e., feedforward PC graphs) on MNIST, FasionMNIST, SVHN, and CIFAR10. The results are compared against popular models in the literature: restricted Boltzmann machines (RBMs) [41], dense associative memories (DAMs) [59], and multilayer perceptrons (MLPs) trained with BP [1]. Classification on MNIST using DAM does not report variance, as it is taken from the original work, and the authors only report the average.

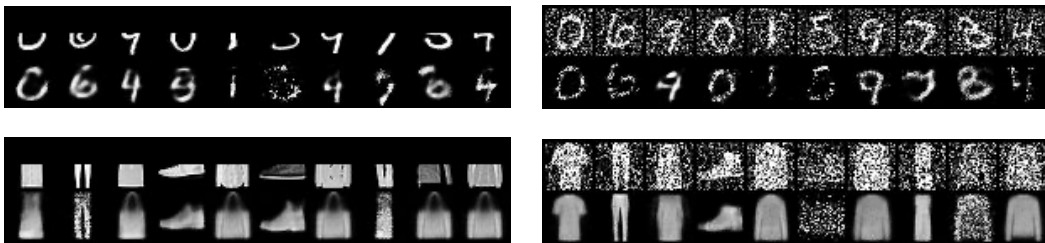

Figure 13: Reconstructed and denoised images using RBMs.

reconstruction is less than 0.001. As corruption, we either removed the top half of the image, or corrupted it with Gaussian noise of mean zero and variance 0.2. The results are shown in Fig. 12.

**Results:** The experiments show that our model is able to well store and retrieve memories, even when tested on colored images. The reconstruction quality, as expected, decreases when adding more memories, and improves when adding more parameters to the model. As hyperparameters, we used $\eta = 0.0001$, $\alpha = 0.5$, and $T = 5$.

## D  Classification Results

In the paper body, we stated that multilayer PCNs are known to perform similarly to BP on classification. Here, we tested this, and compared against popular models in the literature, such as restricted Boltzmann machines (RBMs) [41] and *dense associative memories* (DAMs) [59]. Overall, PCNs are the only models able to perform similarly to BP on the test set. We performed experiments on 4 datasets: MNIST, FashionMNIST, SVHN, and CIFAR10, and the results are in Table 2.

**Setup:** The networks trained using PC and BP have $L = \{2, 3\}$ and 256 hidden neurons each. They are trained using Adam optimization, a weight decay $\lambda \in \{0.001, 0.0001, 0\}$, and the learning rate for the weights $\alpha \in \{0.001, 0.0001\}$. We report the best average results in Table 2. For the RBM, we used a model with 512 hidden nodes, and for the DAM, we copied the official implementation provided by the authors, with the same hyperparameters.

## E  Restricted Boltzmann Machines

To provide a full comparison between the generation capabilities of our model and existing ones in the literature, we trained a different RBM, and performed both reconstructions and denoising tasks. The results are in Fig. 13. Particularly, they show that RBMs sometimes fail to retrieve the correct image, returning a blurry cloud of points in denoising, and tend to often return the same image even when presented with different inputs in reconstruction ones. This problem was consistent in different batches and parametrizations of RBMs, and never happened in any of the models that we have proposed.

**Setup:** We trained several RBMs with $h \in \{256, 512, 1024\}$ hidden nodes, and performed $\{1, 2, 5, 10\}$ *Gibbs samplings*. We always picked the best result.

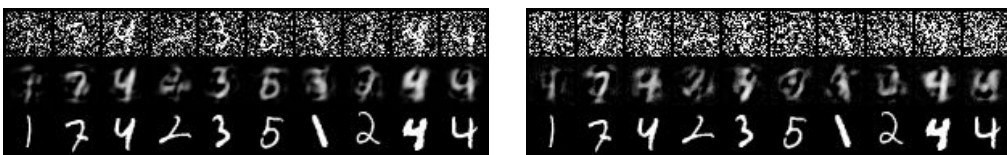

(a)    Predictive Coding Networks with Gaussian noise of variance 0.7 (left) and 1.0 (right).

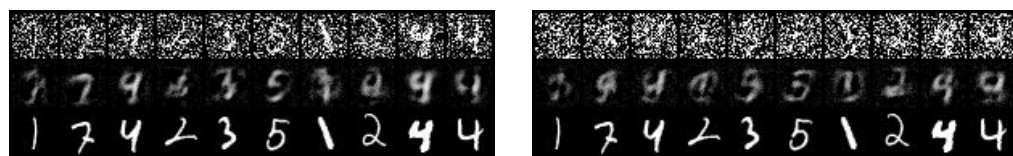

(b)    Autoencoders with Gaussian noise of variance 0.7 (left) and 1.0 (right).

Figure 14: Denoising tasks when presented with high levels of noise.

## F  High Levels of Noise

Here, we push the limits of the model in denoising tasks, where the variance of the Gaussian noise is high enough such that it is often hard for a human evaluator to distinguish different numbers. Particularly, we use a 3 layer PCN with 256 hidden neurons, and we test it against an autoencoder with the same parametrization. The results, provided in Fig. 14 show that both models fail to reconstruct some examples, and the reconstructed ones are noisy. However, we note that PCNs are able to distinguish more numbers than autoencoders, and hence have a better overall performance in this task.

## G  Efficiency of the Model

Training a deep PC network is almost as fast as training deep neural networks with backpropagation. This is despite the fact that every hardware and library is highly optimized for the latter. However, while not faster today, efficiency is an interesting property of PC graphs, and many other neuroscience-inspired learning methods, such as equilibrium and target propagation: all these algorithms are slower than backpropagation; however, they are extremely promising with respect to future developments on the hardware side. In fact, they would allow to train deep neural networks in an end-to-end fashion on physical chips, such as analog circuits [60]. This is something that is not possible to do with backpropagation: in [61], the authors implement exact backpropagation on physical chips. However, the process is quite slow, as there is the need of a digital control signal at every layer of the network. This is due to the sequential structure of deep models, where every operation of a layer has to (1) *wait* for the information of all the previous (following during the backward pass) layers, and (2) be saved in memory via a von-Neumann digital device. The situation would be completely different if using methods that would allow to train neural networks end-to-end, i.e., without any digital component, on the same chip: in this case, the learning process would be much faster, and would not need any external control to be performed. This is possible by using PC. However, despite potential applications on physical chips, PC is also fast on current GPUs, and hence this is not an obstacle towards applications. We now show multiple plots that shot the training and inference times of multiple PC models. Note that these results are obtained by using an implementation that does not make use of the full parallelization capabilities of PC, as this is not supported by standard deep learning frameworks (in our case, Pytorch). Hence, the proposed plots largely overestimate the actual efficiency of PCNs that can be obtained via a correct implementation.

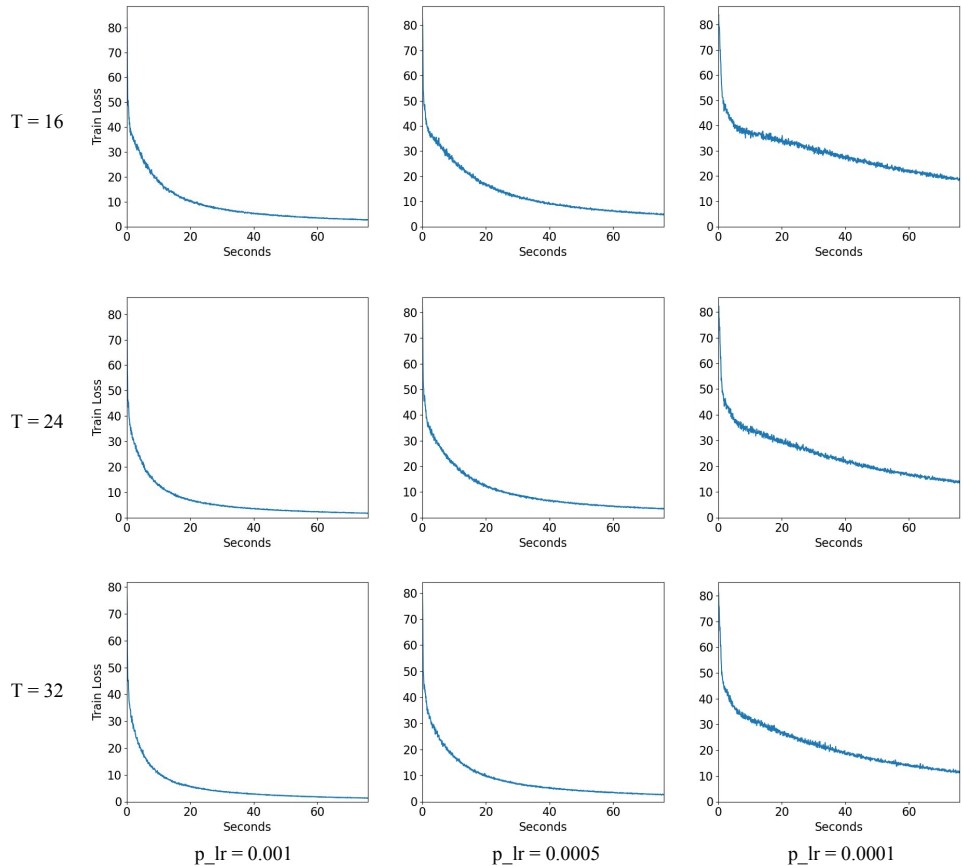

Figure 15: Energy as a function of time (in seconds $s$) for different hyperparameters during training.

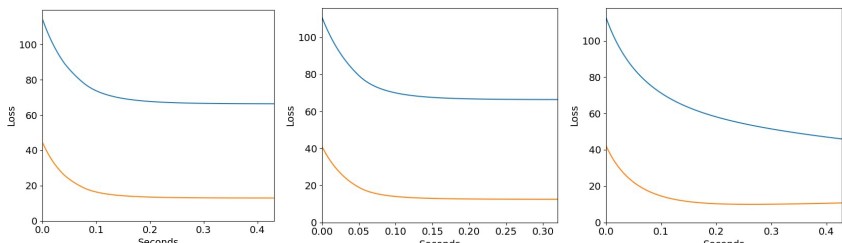

Figure 16: Total energy (blue) and loss (orange) of retrieval (left) and denoising (centre) tasks on a 3-generative model with $512$ hidden neurons per layer. On the right, retrieval of the same model, with added recurrent connections.

**Experiments:** Here, we provide multiple plots that show that PC graphs quickly converge to a stationary point. Particularly, we show that the provided experiments are fast: training a recurrent 3-layer PCN takes about 1 minute on an RTX Titan, as shown in the plots in Fig. 15. Same for testing: reconstructing/denoising an image takes 0.1/0.3 secs, as shown by the plots provided in Fig. 16. Hence, the proposed models are robust to hyperparameter changes and converge rapidly. All the proposed plots are generated via training and testing on a multilayer generative PCN with 3 layers and 512 hidden neurons per layer. We also provide the convergence plot of 48 different PC graphs, of different parametrizations ($N \in \{1500, 2000, 2500, 3000\}$), learning rates ($\alpha \in \{0.0001, 0.00005, 0.00001\}$) and integration steps ($\gamma \in \{1.0, 0.5\}$), on both MNIST and FashionMNIST. As shown in Fig. 17, PC graphs always and quickly converge.

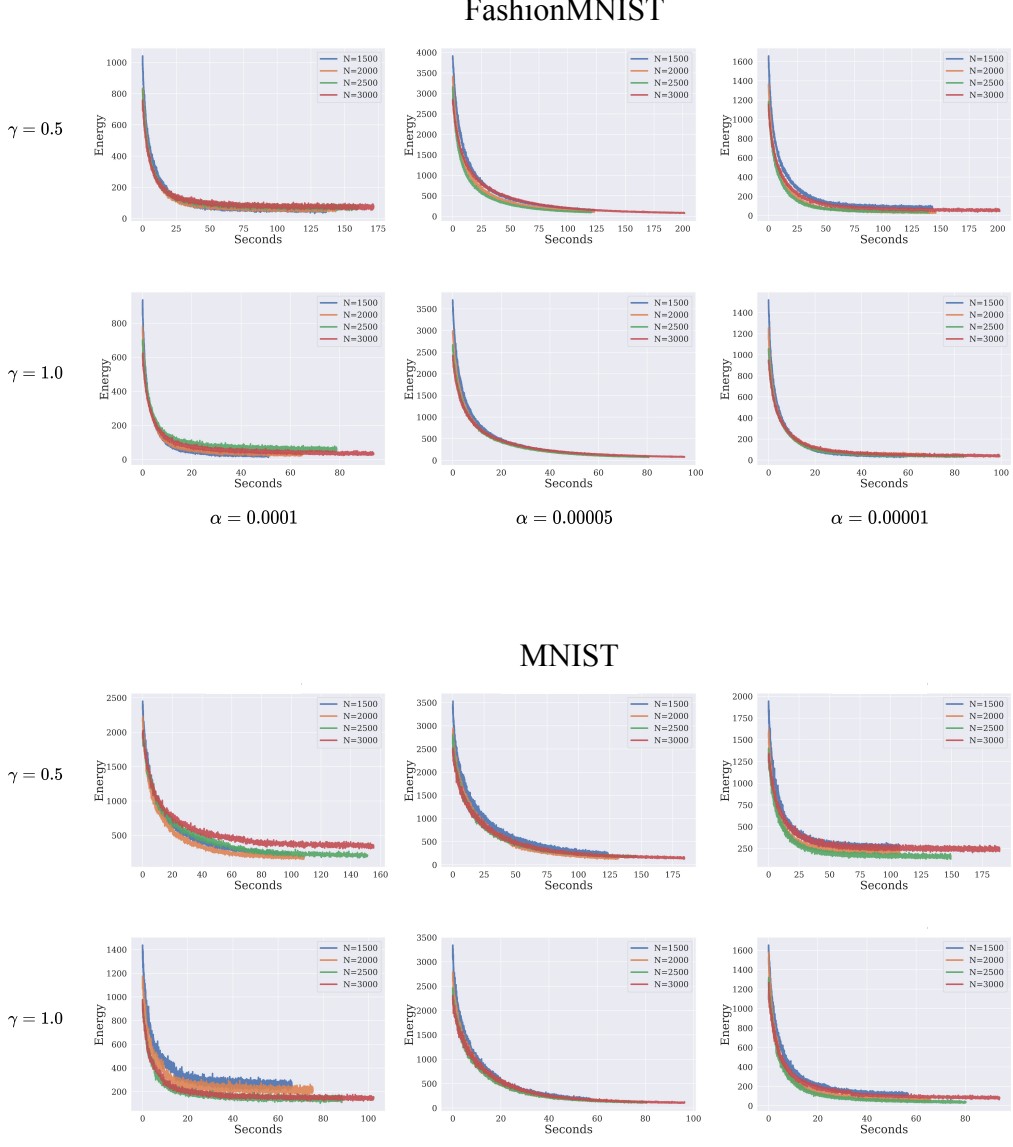

Figure 17: Energy as a function of time (in seconds) for different hyperparameters during training of a PC graph.