# OpenReview forum: "Learning on Arbitrary Graph Topologies via Predictive Coding"
_NeurIPS.cc/2022/Conference — NeurIPS 2022 Accept_

### Official Review · Reviewer_RWvW · 2022-07-01

**Rating:** 7
**Confidence:** 2
**Soundness:** 3 good
**Presentation:** 4 excellent
**Contribution:** 3 good

**Summary:**

The paper describes an approach to training arbitrary neural topologies based on predictive coding (PC). The nodes of the graph corresponding to the inputs are clamped to the observed values, and the remaining nodes are trained locally to minimise the discrepancy between their value and their prediction of the value itself (which is a weighted function of the values of its neighbors). The authors describe the general principle, and then perform a few experiments on MNIST and FashionMNIST showcasing how the model can be used for different applications depending on which nodes are initialised or updated.

**Questions:**

Note: see further comments to the authors for the updated evaluation.

Upon reading several of the papers mentioned in the bibliography, I am unsure about the novelty which is claimed in the paper. In particular, the authors state that they propose an extension to “PC graphs” allowing PC to be run on arbitrary graph topologies, but this was already done in the majority of cited papers, including:

https://proceedings.neurips.cc/paper/2021/hash/1fb36c4ccf88f7e67ead155496f02338-Abstract.html

The above paper also provides examples about using the network as an associate memory, which encompasses half of the experiments done in this paper. The algorithm which is used is also equivalent (gradient descent for T steps on values and weights after fixing a single input example). Since this is my major concern, I am currently rating this as a borderline paper, as it appears to be a collection of previous known results.

A few additional questions I have:

•	The authors should be more comprehensive in describing the comparisons of their method with Boltzmann machines and Hopfield networks, since they share a number of common characteristics.
•	The convergence guarantees of the algorithms wrt the loss (sum of local errors squared) is not discussed.
•	In general, how efficient is the training compared to a standard backpropagation?

**Strengths And Weaknesses:**

Note: see further comments to the authors for the updated evaluation.

Let me premise that I am not familiar with the literature on PC and its applications to deep learning, and I have learned about most papers by following the references listed in the paper itself. The paper is well written, and the authors makes a good case for the strengths of the approach (locality) and in particular its neuroscience background. The experiments are interesting, and all the diagrams in the paper are very clear (e.g., how different masks over the connectivity can results in different known formulations).+

Experiments are done on very simple CV datasets with no comparisons. However, I think this is expected of papers that are trying to modify the way neural networks are trained at a fundamental level. Still, the authors are not even mentioning (or comparing) to the literature they are citing on PC, which is strange.

---

> ### Author Response · Authors · 2022-08-02
> **Answer**
>
> We thank the reviewer for his time and comments.
>
> > Q:  I am unsure about the novelty which is claimed in the paper. In particular, the authors state that they propose an extension to “PC graphs” allowing PC to be run on arbitrary graph topologies, but this was already done in the majority of cited papers.
>
> The majority of the cited papers (all the ones that use PC to train deep neural networks), perform experiments using **hierarchical structures only**. Including Salvatori et al.,2021  the one you have cited. To our knowledge, no one has tested PC to the level of generality we have done in this paper.
>
> > Q: Half of the experiments have already been done in the associative memory paper you have cited.
>
> This is incorrect: the work you cite performs **associative memory** experiments: the reconstructions and denoising tasks are done on **training images**, while the vast majority of the experiments that we do in this paper are done on **test images**, i.e., images the network has never seen before.
> There is only one associative memory experiment in our work: the one reported in Figure 3(e). This is indeed conceptually equivalent to the ones performed in Salvatori et al. However, the difference is that here we use a fully connected graph, while Salvatori et al. use hierarchical networks. This is in line with all the experiments that we have performed in Figure 3: they have all been done before, just not on fully connected graphs. Exploiting the properties of PC on graphs of any topology is the goal of this paper. All in all, there is no intersection in the results between the work you have cited and ours, as we only perform one associative memory experiment, but we do it on a graph with different topology.
>
> > Q: The algorithm which is used is also equivalent (gradient descent for T steps on values and weights after fixing a single input example).
>
> Note that we never claim to have invented the algorithm: all the merit goes to Rao and Ballard, 1999. All the papers that tackle machine learning problems with predictive coding use the same (or variations of) algorithm in the same way most classical machine learning papers use backpropagation. We claim, however, to have shown how this algorithm can be used to train graphs of any topology, and hence be able to perform tasks that could not be possible to perform on feedforward architectures, such as performing all the experiments in Fig. 3 (a,b,c,d) + classification after training only one architecture.
>
> > Q: The authors should be more comprehensive in describing the comparisons of their method with Boltzmann machines and Hopfield networks, since they share a number of common characteristics.
>
> We have added such discussion in the related works. Particularly, we have highlighted the following differences:
> 1) Boltzmann machines and Hopfield networks are pure Hebbian learning methods, as they do not encode the concept of error, fundamental in PC and probably key to its success;
>
> 2) As a consequence of this, Boltzmann machines and Hopfield networks only have undirected connections, while every synapse in PC is directed.
>
> 3) While both are energy-based methods, that infer via a relaxation process of a physical system, Boltzmann machines and Hopfield networks are simply initialized with a test point (i.e., they do “query by initialization”). PC, on the other hand, can also do ‘query by conditioning’. Actually, this is by far the most used query in practice, and the most effective.
>
> > Q: The convergence guarantees of the algorithms wrt the loss (sum of local errors squared) is not discussed.
>
> In the supplementary material, we have added the convergence plots of over 40 PC graphs, trained on different datasets and with different parameters, which show that training on PC graphs always converges empirically. This can be found in the last figure of the supplementary material, Fig.15, with a discussion in the same section.
> Theoretical convergence guarantees, however, cannot be provided, in the same way they are hardly provided by deep neural networks trained with backpropagation: the complexity and non-linearity of the model make the study intractable. However, we believe that we cannot be penalized by the lack of this theoretical result: the whole machine learning community has been struggling to provide such a result in the general case for over 30 years, and all the available results are about extremely overparametrized networks.

---

> > ### Comment · Reviewer_RWvW · 2022-08-04
> > **Answer to the authors**
> >
> > I thank the authors for the rebuttal, and I apologize about some mistakes in my original review which, as stated, stems from my lack of knowledge of this area. Upon re-reading the paper and the bibliography, I agree with the assessment by the authors and I am happy to raise my evaluation to an acceptance.

---

### Official Review · Reviewer_v3jK · 2022-07-08

**Rating:** 5
**Confidence:** 3
**Soundness:** 3 good
**Presentation:** 3 good
**Contribution:** 2 fair

**Summary:**

This paper presents the use of predictive coding (PC) to perform various learning and inference tasks. In particular, the paper proposes PC graphs that generalize PC to arbitrary network structures. The flexibility of PC graphs is then demonstrated experimentally using a variety of network structures and learning tasks.

**Questions:**

- Corruption noise variance values of 0.2 and 0.5 are quite small. How would a larger choice affect the analysis? Would the method still work in that case?
- P. 6, line 211: Missing SVHN dataset

**Limitations:**

The authors do not discuss the limitations and societal impact of their work.

**Strengths And Weaknesses:**

- Originality: While the idea of PC graphs in the paper is not new, this paper however focuses on showing how a single PC graph model can be extensively used in various ways to learn different tasks.

- Quality: The PC graph model is technically valid though calling it a biologically plausible model seems slightly misleading since the existence of error nodes is biologically questionable. The claim for the flexibility of the model is well supported by the experiments on various learning settings.

- Clarity: The paper is clearly written and well-organized.

- Significance: While the paper does not advance state-of-the-art results, the paper clearly presents the flexibility of PC graphs in various learning tasks through experiments. However, the overall practicality/utility of the model is still hard for readers to assess since 1) the potential computational overhead of the method is only briefly illustrated in the supplementary material, which is not convincing and 2) experimental results are mostly qualitative with some quantitative comparison against a few simple baselines. Additional knowledge of its computational efficiency and demonstrated effectiveness would instill more trust into the method.

---

> ### Author Response · Authors · 2022-08-02
> **Answer**
>
> We thank the reviewer for his time and comments.
>
> > Q: The idea of PC graph is not new:
>
> Predictive coding as a learning algorithm is not new: the original formulation was developed by Rao and Ballard, 1999. It has in fact already been applied to hierarchical structures in multiple previous works. However, to our knowledge, we are the first to develop the general idea of PC graphs, showing and testing how predictive coding can be trained on neural networks with cycles and any graph topology, extensively testing different properties and different architectures, such as the assembly of neurons.
>
> > Q: Calling PC a biologically plausible model seems slightly misleading, since the existence of error nodes is biologically questionable.
>
> We agree that research about the existence of error nodes has so far been inconcludent, and hence still to be proven. However,  there are two reasons that have motivated us to adopt the terminology anyway: (1) calling PCNs ‘biologically plausible’ is common practice in the field, and (2) PCNs can also be mapped on a different neural architecture in which errors are encoded in apical dendrites rather than separate neurons [1]. Such mapping of PCN algorithm on network with errors in dendrites has been described in Box 4 of [2].
>
> We have added the above discussion, as well as a figure explaining the differences between the two implementations (Figure 7, beginning of supplementary material), by making clear what we mean by “biologically plausible'', by addressing your concern about error neurons. Particularly, we have written the following:
>
> *In the literature, there is often a disagreement on when a specific algorithm can be considered biologically plausible. This follows, as every computer simulation fails to be completely equivalent to every aspect on how the brain works, as there will always be some details that make the simulation implausible. Hence, it is normally assumed that an algorithm is biologically plausible when it satisfies a list of properties that are also satisfied in the brain. Different works consider different properties. In our case, we consider as list of minimal properties that a learning rule should satisfy, the ones that allow to have a possible neural implementation, such as local computations and lack of a global control signal to trigger the operations. However, the neural implementation proposed in Fig.2 takes error nodes into account, often considered implausible from the biological perspective [1]. Even so, the biological plausibility of our model is not affected by this: it is in fact possible to map PCNs on a different neural architecture, in which errors are encoded in apical dendrites rather than separate neurons [1,2]. Graphical representations of the differences between the two implementations can be found in Fig. 7, taken (and adapted) from [2].*
>
> [1] Sacramento, João, et al. "Dendritic cortical microcircuits approximate the backpropagation algorithm." Advances in neural information processing systems 31 (2018).
> [2] Whittington, James CR, and Rafal Bogacz. "Theories of error back-propagation in the brain." Trends in cognitive sciences23.3 (2019): 235-250.
>
>
> > Q: The potential computational overhead of the method is only briefly illustrated in the supplementary material, which is not convincing.
>
> Every model proposed in the paper is extremely fast to train and test. However, an experimental analysis of the training\convergence speed of PC graphs has not been provided. We do it now: we have performed multiple experiments, and show the convergence speed of multiple models under multiple parameters. They all converge in less than two minutes (most of them, in less than one). We have added the plots that show the training speed of 48 models (all of our hyper parameter search). We have added the plots in the last figure of the supplementary material, Fig.15, with a discussion in the same section. Please let us know in case you need any further clarifications.
>
> > Q: The experimental results are mostly qualitative with some quantitative comparison against a few simple baselines.
>
> We are performing the comparison according to multiple metrics, and will report all the results here and in the final version of the paper.
>
> > Q: Corruption noise variance values of 0.2 and 0.5 are quite small. How would a larger choice affect the analysis? Would the method still work in that case?
>
> Yes, it would, but the performance would be worse. In the supplementary materials (Section E), we have added examples of denoised images when providing Gaussian noise with variance = 0.7 and 1.0. Note that the examples that are badly retrieved are extremely hard to distinguish even from the human eye perspective, and hence we cannot expect the model to work well on these. We have also attached equivalent experiments performed with autoencoders with the same parametrization (depth/width). As it is easy to see, the reconstructions of PCNs are better

---

> ### Author Response · Authors · 2022-08-08
> **FID Scores**
>
> We have now added the FID score on the reconstructions on MNIST. Particularly, we have used the same networks used to generate the corrupted images of Figure 5. We did them according to the official implementation:
>
> https://github.com/abdulfatir/gan-metrics-pytorch
>
> As you can see from the results reported in Figure 7 of the updated paper, PC constantly outperforms auto encoders with the same structure and parametrisation. Particularly, the best FID scores obtained by PC networks on images corrupted with Gaussian noise of variance 0.2, 0.5, 0.7 are  25.6, 44.53, and  51.38, respectively. For BP, we have a much worse 43.93, 53.79, and 57.56.  (a large hyper parameter search was performed in both cases).
>
>  We thank the reviewer for the pointer, as we believe this experiment has improved the quality fo the paper. We  have addressed the two experimental weaknesses pointed out (FID score and efficiency) by performing the requested experiments. We kindly ask the reviewer to increase the score if he believes the arguments we brought up are reasonable.

---

> > ### Comment · Reviewer_v3jK · 2022-08-09
> > **Reply to Authors**
> >
> > Thank you for the clarifications and additional experiments. The clarifications are much appreciated while the experiments alleviate some of my concerns. After reading and considering the author response, I will still maintain my original score.

---

### Official Review · Reviewer_5yBi · 2022-07-12

**Rating:** 5
**Confidence:** 3
**Soundness:** 2 fair
**Presentation:** 4 excellent
**Contribution:** 2 fair

**Summary:**

This paper proposes a new framework, PC graphs, which can train on arbitrary network structures without the limitation of backpropagation.
It is inspired by neuroscience and implemented by predictive coding. The authors show how to use this framework to solve different tasks, e.g., classification, generation, reconstruction and denoising.

**Questions:**

1. From the accuracy of MNIST, the performance of PC graphs is still a issue, which needs to be improved. The authors should show the audience its potential ability to surpass artificial neural networks.
2. The training time of PC graphs, or convergence situation of different graphs, is not mentioned in the paper.
3. Could author use some quantitative evaluation metrics to show the performance of PC graphs in generation, reconstruction and denoising tasks?

**Limitations:**

not available

**Strengths And Weaknesses:**

Strengths:
1. It is very interesting to leverage biological insights to design a new training framework.
2. The authors analyze how this framework is applied to solve different tasks, and the relationships between classical neural networks, like the feedforward network.
3. The paper is well-written and easy to follow.


Weaknesses:
1. The theoretical analysis of PC graphs is missing, such as time complexity, speed of convergence, and so on.
2. The experiments are limited in some datasets, or qualitative. The accuracy of MNIST is ~92%, which is much lower than simple CNNs.

---

> ### Author Response · Authors · 2022-08-02
> **Comment to the weaknesses**
>
> We thank the reviewer for his time and comments.
> > Q:  The performance of PC graphs is still an issue, which needs to be improved. The authors should show the audience its potential ability to surpass artificial neural networks. CNNs perform much better.
>
> All the experiments on the fully connected PC graphs (Section 3, Experiments on Fully Connected Graphs) are a **proof of concept** of our method, the goal is not to show how well they do on classification tasks. The goal is instead to show that it is possible to train models able to perform any kind of task (generation, demonising, image completion, associative memory, classification, conditional generation, etc.) on the data, without the need of training the model on different ones. They can be seen as toy experiments, where we have used the most general, and hence inefficient structure.
>
> We agree that CNNs have a better classification accuracy: they have hierarchies, convolutions, gradient sharing, and are focused and optimized to perform only one task: classification. However, we should not confront CNNs against our model, a general proof-of-concept method with an extremely simple structure, without any hierarchy, which is not focused on any specific task, but instead aims to learn an internal representation of the dataset that can be queried in any way.
>
> However, please do note that PC graphs do work very well on practical tasks when the graph has a more specific shape: we simply have to change the architecture used. To achieve better results (not only in classification tasks, but also in generation), for example, hierarchy is fundamental. In fact, an obstacle towards obtaining results that are comparable with the ones of backpropagation is the structure: as soon as we remove specific connections and make the network hierarchical, our performances are equivalent to those of BP, as shown in Table 2 in the supplementary material. Furthermore, when adding a convolutional structure, PCNs are able to obtain reasonable performance on complex datasets, such as ImageNet [1].
>
> In summary, the low performance of PC graphs with the most general structure in classification tasks should not be considered as a weakness of the paper, and we hope to have brought enough arguments to convince the reviewer of this.
>
> [1] Han, K., Wen, H., Zhang, Y., Fu, D., Culurciello, E., & Liu, Z. (2018). Deep predictive coding network with local recurrent processing for object recognition. Advances in Neural Information Processing Systems, 31.
>
> > Q: The training time of PC graphs, or convergence situation of different graphs, is not mentioned in the paper.
>
> We have added an analysis of the convergence speed of over 40 models, under multiple parametrizations (last plot of the supplementary material, Fig.15, with a discussion in the same section). In short, PC graphs are extremely fast to train, and have always converged to a solution under every single parametrization used. We have added this in the revised paper. Here is a sketch of the section:
> Training a PC graph is extremely fast. This is due to the fact that it requires a small value of $T$, and hence a small number of non-parallel matrix multiplications to be performed. The training time of all the proposed experiments was below two minutes (most of them, below one minute), as we show in the plots in the linked figure. We also present multiple plots concerning the convergence when multiple parameters are changed.
>
> We hope to have extensively addressed the two weaknesses you have pointed out. Please let us know in case you need any further clarifications.
>
> > Q: Metrics for generation tasks
>
> We are performing the comparison according to multiple metrics, and will report the results here and in the final version of the paper when ready.

---

> > ### Author Response · Authors · 2022-08-08
> > **FID Scores**
> >
> > We have now added the FID score on the reconstructions on MNIST. Particularly, we have used the same networks used to generate the corrupted images of Figure 5. We did them according to the official implementation:
> >
> > https://github.com/abdulfatir/gan-metrics-pytorch
> >
> > As you can see from the results reported in Figure 7 of the updated paper, PC constantly outperforms auto encoders with the same structure and parametrisation. Particularly, the best FID scores obtained by PC networks on images corrupted with Gaussian noise of variance 0.2, 0.5, 0.7 are  25.6, 44.53, and  51.38, respectively. For BP, we have a much worse 43.93, 53.79, and 57.56.  (a large hyper parameter search was performed in both cases).
> >
> >  We thank the reviewer for the pointer, as we believe this experiment has improved the quality fo the paper. We  have addressed the two experimental weaknesses pointed out (FID score and efficiency) by performing the requested experiments, and largely discussed the third one (why accuracy on PC graphs is low). We kindly ask the reviewer to increase the score if he believes the arguments we brought up are reasonable.

---

> > > ### Comment · Reviewer_5yBi · 2022-08-09
> > > **Reply to Authors**
> > >
> > >  I would like to thank the authors for their reply. I have read all the rebuttal and the modified paper.
> > >
> > >  Although the authors still do not address my concern about the potential performance of PC graphs, I appreciate that the authors show flexibility in different tasks.
> > >
> > >  I increase my score to 5.

---

> > > > ### Author Response · Authors · 2022-08-09
> > > > **Performance of PC Graphs**
> > > >
> > > > We thank the reviewer for taking time to address our rebuttal.
> > > >
> > > > We would like to further stress that the low performance are only obtained by **fully-connected** PC Graphs, not PC Graphs in general. This is proven in the supplementary material, where we show that PC Graphs with **hierarchical structures** are able to reach competitive (and sometimes slightly better) performance than back propagation. Hence, the claim that PC graphs always obtain bad classification performance is incorrect, as shown in Table 2 of the supplementary material. It is true only in the toy example we have provided for the sake of a good presentation.
> > > >
> > > > Note that the toy example on fully connected graphs was only introduced to show how our method works. It helps to reach our goal of successfully presenting our work, research and results: Indeed, all reviewers stated that the quality of the presentation is **excellent**. This was possible also thanks to a step-by-step presentation that (1) introduced the main equations of the model, (2) provides toy examples on how the model works even in the simplest case, and (3) shows the performance of more complex models. It feels unjust to be punished by the bad results of these toy examples mentioned in (2), that by no means were meant to be judged in terms of performance.

---

> > > > > ### Comment · Reviewer_5yBi · 2022-08-10
> > > > > **Reply**
> > > > >
> > > > > Thanks for your reply. I checked the performance in the supplementary material. I found an experimental result:
> > > > > | Model | PC | RBM | DAM | BP |
> > > > > | --- | --- |  --- | --- | --- |
> > > > > | CIFAR10 | 56.23 ± 3.36 | 41.12 ± 3.88 | 46.06 ± 2.77 | 59.11 ± 2.47 |
> > > > > where PC is the proposed method and BP is MLPs trained with BP. But the test accuracy of a simple Reset is >80%. Thus, I have a concern about performance.
> > > > >
> > > > > Besides, these CV datasets are small and Imagenet is the current benchmark. Compared with MNIST and CIFAR10, I believe there is a significant realistic meaning when the accuracy is high on Imagenet. Otherwise, it is still a toy model which has the potential to be SOTA.

---

> > > > > > ### Author Response · Authors · 2022-08-10
> > > > > > **Comparison against BP**
> > > > > >
> > > > > > Note that  the performance of the reported tables are obtained with MLPs and without data augmentation. This explains the bad performance of both BP and PC: they would both reach (and surpass) the 80% of ResNets that you mentioned with the correct architecture. I can provide two references that show the potential of PC to match the performance of BP on much more complex tasks, such as ImageNET:
> > > > > >
> > > > > > In [1], the authors show that a variation of PC well performs on ImageNet (table 4, page 7), and obtains results much higher than the 80% you mention on CIFAR10 in Table 2. Note that the training algorithm is basically the same.
> > > > > >
> > > > > > In [2], the authors show that by simply adding some external control to PC (meaning, simply telling which weights to update at which time step during the inference process), it is possible  to **exactly** replicate the weight update of BP on any given architecture. This means that PC can theoretically be used to train models such as GPT3 with good performance. This is a formal result.
> > > > > >
> > > > > > All in all, we agree with the reviewer: when the only goal is to have a good classifier that obtains very good performance on a single task, such as classification on ImageNet, we would not use a PC Graph, but a model similar to that of [1]. However, if the need is to have a more flexible model, where we do not have prior information on which task it will be used, PC graphs will be useful. In summary, the goal of this paper focuses on scenarios where a much more flexible architecture to be queried in different ways is needed.
> > > > > >
> > > > > >
> > > > > > [1] Han, K., Wen, H., Zhang, Y., Fu, D., Culurciello, E., & Liu, Z. (2018). Deep predictive coding network with local recurrent processing for object recognition. Advances in Neural Information Processing Systems, 31.
> > > > > >
> > > > > > [2] Salvatori, Tommaso, et al. "Reverse Differentiation via Predictive Coding." Proceedings of the 36th AAAI Conference on Artificial Intelligence ‚AAAI 2022 ‚Vancouver, BC, Canada, February 22--March 1 ‚2022. Vol. 10177. AAAI Press, 2022. (edited)

---

> > > > > > > ### Comment · Reviewer_5yBi · 2022-08-10
> > > > > > > **Reply**
> > > > > > >
> > > > > > > Thanks for providing two useful references. Now it is clear for me to understand the potential ability of PC graphs from different aspects, like accuracy and flexibility.

---

> > > > > > > > ### Author Response · Authors · 2022-08-10
> > > > > > > > **Reply**
> > > > > > > >
> > > > > > > > We thank the reviewer for the final comment. If you are satisfied with our response, we would appreciate it if you could update your score to account for the explanation we have provided above. We will include the above discussion in the final version of the manuscript.

---

### Author Response · Authors · 2022-08-08
**Overall changes**

I would also like to add this as a general comment to all the reviewers (as they are ignoring us and not answering):
We thank the reviewers for the time spent reviewing our paper, as well for all the suggestions that made us improve our current submission. Particularly, we have introduced the following changes, that have addressed the concerns and questions raised in the reviews so far:

1) We have added a section in the supplementary material (last figure/section), where we discuss the efficiency of PC Graphs. This was done to address what was pointed out by reviewers 5yBi and v3jK, that pointed out that we did not discuss the efficiency/time complexity/ speed of our models. More in detail addressed this by providing **48** plots of the total energies as functions of time (seconds) for PC graphs. We believe this addresses the concern.

2) We have now added the FID score on the reconstructions on MNIST. Particularly, we have used the same networks used to generate the corrupted images of Figure 5. We did them according to the official implementation:

https://github.com/abdulfatir/gan-metrics-pytorch

As you can see from the results reported in Figure 7 of the updated paper, PC constantly outperforms auto encoders with the same structure and parametrisation. Particularly, the best FID scores obtained by PC networks on images corrupted with Gaussian noise of variance 0.2, 0.5, 0.7 are 25.6, 44.53, and 51.38, respectively. For BP, we have a much worse 43.93, 53.79, and 57.56. (a large hyper parameter search was performed in both cases). This was done to address what was pointed out by reviewers 5yBi and v3jK, and we believe this addresses the concerns.


3) We have added a detailed discussion about the biological plausibility of our method (first section of the supplementary material). Particularly, we have stated that PCNs can also be mapped on a different neural architecture in which errors are encoded in apical dendrites rather than separate neurons [1]. Such mapping of PCN algorithm on network with errors in dendrites has been described in Box 4 of [2]. We have added the above discussion, as well as a figure explaining the differences between the two implementations (Figure 7, beginning of supplementary material), by making clear what we mean by “biologically plausible'', by addressing your concern about error neurons. This was done to address the concern raised by Reviewer v3jK.


We have also addressed minor comments by all the reviewer, detailed in the individual responses. Overall,  if you are satisfied with our response, we would appreciate it if you would update your score to account for the changes we have made to the manuscript. Otherwise, as the time for rebuttals is running out, we kindly ask the reviewers to point out any other remaining concerns, question or suggestion. We would be pleased to address those points.

---

### Meta-Review · Area_Chair_kLah · 2022-09-01

**Recommendation:** Accept
**Confidence:** Certain

**Metareview:**

This work presents the use of predictive coding (specifically PC graph) as a way to overcome some of the limitations of the commonly used backpropagation approach in deep learning. The initial reviews have raised some concerns regarding the paper, but these were sufficiently addressed in rebuttal to result in one reviewer recommending acceptance and others leaning towards acceptance. Taking this into account, and since I also think this line of work would contribute well to the audience of NeurIPS, I recommend acceptance and I would like to encourage the authors to take into account the reviewer comments, and the points discussed in their rebuttal, when preparing the camera ready version of the paper.

**Award:**

No

---

### Decision · Program_Chairs · 2022-09-14

Accept